# Hypersensitivity to Intravenous Iron Preparations

**DOI:** 10.3390/children9101473

**Published:** 2022-09-27

**Authors:** Silvia Caimmi, Giuseppe Crisafulli, Fabrizio Franceschini, Lucia Liotti, Annamaria Bianchi, Paolo Bottau, Francesca Mori, Paolo Triggiano, Claudia Paglialunga, Francesca Saretta, Arianna Giannetti, Giampaolo Ricci, Carlo Caffarelli

**Affiliations:** 1UOC Pediatria, Fondazione IRCCS Policlinico San Matteo, 27100 Pavia, Italy; 2UOC Pediatria, Università di Messina, 98124 Messina, Italy; 3UOC Pediatria, Azienda Ospedaliero-Universitaria “Ospedali Riuniti”, 60020 Ancona, Italy; 4UOC Pediatria, Azienda Ospedaliera San Camillo Forlanini, 00152 Roma, Italy; 5Dipartimento di Pediatria e Neonatologia, Ospedale di Imola, 40026 Imola, Italy; 6Allergy Unit, Meyer Children’s Hospital, 50139 Florence, Italy; 7UOC di Pediatria, Azienda Ospedaliera-Universitaria “Consorziale-Policlinico”, Ospedale Pediatrico Giovanni XXIII, 70123 Bari, Italy; 8SC Pediatria, Ospedale Latisana-Palmanova, Dipartimento Materno-Infantile Azienda Sanitaria Universitaria Friuli Centrale, 33100 Udine, Italy; 9Pediatric Unit, IRCCS Azienda Ospedaliero-Universitaria di Bologna, 40138 Bologna, Italy; 10Department of Medical and Surgical Sciences (DIMEC), University of Bologna, 40138 Bologna, Italy; 11Clinica Pediatrica, Azienda Ospedaliero-Universitaria, Dipartimento Medicina e Chirurgia, Università di Parma, 43126 Parma, Italy

**Keywords:** iron, iron hypersensitivity, desensitization, drug allergy, hypersensitivity reaction, drug provocation test

## Abstract

Intravenous iron is widely used for the treatment of iron deficiency anemia when adherence to oral iron replacement is poor. Acute hypersensitivity reactions during iron infusions are very rare but can be life threatening. Major risk factors for hypersensitivity reactions include a previous reaction to an iron infusion, a fast iron infusion rate, multiple drug allergies, atopic diseases, high serum tryptase levels, asthma, and urticaria. The management of iron infusions requires meticulous observation, and, in the event of an adverse reaction, prompt recognition and severity-related interventions by well-trained medical and nursing staff. Avoidance of IV iron products in patients with iron hypersensitivity reactions may not be considered as a standard practice.

## 1. Introduction

Iron-deficiency anemia is a global health problem that is associated with a wide breadth of underlying conditions including malabsorption, poor diet, feeding difficulties, chronic hidden bleeding, chronic kidney disease, heart failure, underlying inflammatory conditions, cancer, chronic intestinal inflammatory disease, surgery and the perinatal period. Oral iron replacement is the standard therapy in iron-deficiency anemia. However, adherence is hampered by poor taste, long-term treatment, impediment to oral ingestion, insufficient absorption and gastrointestinal symptoms, such as constipation, upper digestive tract irritation or staining of teeth, found in 59% of patients [1]. The intramuscular (IM) route has been considered safer than the intravenous (IV) route, but published data do not support this view, since the two routes of administration show a similar incidence of reported adverse events (AEs). However, IM injections need numerous doses, are painful, cause permanent discoloration of the skin and have been associated with gluteal sarcomas at the injection area. Therefore, the IM route has been abandoned in favor of the IV route [1]. IV formulations may be the first option when iron deficiency needs to be quickly corrected with a rapid refill of iron storages. IV high-molecular-weight iron dextran (HMWID) formulations have been associated with an unacceptable incidence of severe immediate hypersensitivity reactions (HSRs), raising public concern, and they were, therefore, withdrawn from the market. Current marketed compounds have proven to be safer, although it has been documented that they can also elicit HSRs (Table 1).

The safety of these compounds is an issue that has been rendered sensitive because they are frequently used. In this review, we describe the epidemiology, pathogenetic mechanisms, clinical features and management of HSRs to the available IV iron products.

## 2. Frequency of Hypersensitivity Reactions to Intravenous Iron Formulations

Following the withdrawal of HMWID, IV iron therapies are associated with an estimated severe AE incidence of less than 1 in 250,000 doses [4] and rarely with anaphylaxis [5,6,7,8]. Population-based studies in the United States have reported that the risk for anaphylaxis was 4.0 to 6.8 per 10,000 first IV iron dextran administrations and 2.0 to 2.4 per 10,000 first IV iron non-dextran administrations [9]. Population studies in Europe are lacking. From 1 January 2014 to 31 December 2019, the U.S. Food and Drug Administration Adverse Event Reporting System database, constructed on post-authorization spontaneous reports, registered 57 HSRs and 22 anaphylactic reactions with 4 deaths to IV iron dextrans; 212 HSRs and 43 anaphylactic reactions with 7 deaths to IV iron sucrose; 196 HSRs and 67 anaphylactic reactions with 23 deaths to IV ferumoxytol; 401 HSRs and 51 anaphylactic reactions with 1 death to IV ferric carboxymaltose [10] (Table 2).

According to insurance claims from Medicare (U.S.) in American older adults, the adjusted incidence rate of anaphylaxis was retrospectively reported to be 9.8 in 10,000 first IV administrations of low-molecular-weight iron dextran, 4.0 for ferumoxytol, 1.5 for ferric gluconate, 1.2 for iron sucrose and 0.8 for ferric carboxymaltose [11]. A post-marketing safety study in European adults reported lower figures. The incidence proportion of anaphylaxis ranged from 0.38 to 0.51 per 10,000 first IV iron doses and from 0.44 to 0.55 for iron non-dextrans; the incidence of anaphylaxis was undetermined for iron dextran formulations as no event occurred [5]. In children, second-generation products (i.e., iron sucrose, ferric gluconate and low-molecular-weight iron dextran) have been rarely associated with serious AEs and are currently used. Regarding this, in a retrospective analysis performed on 38 children treated with iron sucrose between 2004 and 2009, receiving a total of 510 doses of iron sucrose, 6 AEs were registered, with only one being severe. This latter instance occurred in a female weighing 57 kg with a dose of 500 mg, greater than recommended [12]. In 2017, a retrospective study in 142 children and young adults aged 22 years and below receiving IV iron sucrose over a ten-year period, reported one case of infusion interruption due to cough and wheezing at the beginning of the administration. The patient recovered promptly after stopping the infusion [13]. Finally, no infusion-associated severe AEs were reported over a 6-year period among 194 patients aged 21 years or younger treated with a combined 1088 doses administered (including iron sucrose, ferric gluconate and low-molecular-weight iron dextran) [14]. Regarding third-generation IV iron formulations (i.e., ferumoxytol, ferric carboxymaltose (FCM) and iron isomaltoside), there is a paucity of safety data in children since they have not been approved for children <14 years old in Europe and <18 years old in the U.S. [15], apart from FCM, only recently approved for children >1 year old in the U.S. [16]. However, over the period 2014–2021, a systematic review found 11 studies with 866 children receiving 1231 FCM infusions [17]. AEs to FCM were reported in 52 (6%) children, with degrees of severity between I and III according to the “Common terminology Criteria for AE” [15]. Reactions included skin rash, pruritus, mild urticaria, dizziness, mild fever, headache and extravasation injury and one case of mild anaphylactic reaction with a transient drop in oxygen saturation [17]. Interestingly, 147 children <6 years old had milder reactions (grade I and II) [17]. A complete allergy work up was not performed. Current data indicate a positive risk–benefit ratio and a low risk that IV iron formulations induce HSRs or anaphylaxis, which can be life threatening when not rapidly treated [8]. Therefore, the Committee for Medicinal Products for Human Use of the European Medicine Agency (EMA-CHMP) recommended that IV iron infusions need to be administered only in facilities with qualified personnel able to manage anaphylactic reactions and where resuscitation equipment is immediately available [18].

## 3. Mechanisms of Reactions to Intravenous Iron Infusion

IV iron compounds are colloidal solutions based on small spheroidal micellae with an iron hydroxide core gel and an envelope formed by iron–carbohydrate particles (Figure 1). Products differ from each other by the size of their core and the identity and density of the surrounding carbohydrates. Carbohydrates stabilize the gel, slow the release of iron and maintain the resulting particles in colloid suspension. The rate of release of bioactive iron is inversely related to the strengths of the complex: stronger complexes have a lower potential to saturate transferrin with subsequent lower free iron toxicity [1,2,19]. 

Several hypotheses have been offered to explain the underlying mechanisms of HSRs to IV iron administration. The mechanisms of drug HSRs are often categorized in the classical scheme of Gell and Coombs, which distinguishes four types (I–IV) of HSRs. No evidence is available that HSRs are IgE-mediated reactions. There are some data that HSRs to HMWID may be mediated by IgG antibodies to dextran (immune complex reaction—Type III) that had been produced before exposure to HMWID [7]. Along this line, it is noteworthy that specific IgG and IgM antibodies to dextrans used as plasma expanders or anticoagulants have been detected [16] and IgG antidextran binds the dextran-derived nanoparticles ferumoxytol and ferric derisomaltose [20,21]. Circulating IgG antibodies against carbohydrates nanoparticles, such as ferric carboxymaltose and iron sucrose, have not been detected following the use of non-dextran iron formulations [20,22]. 

Besides Gell and Coombs mechanisms, complement-activation-related pseudo-allergy (CARPA) may be involved. CARPA is a non-immune, antibody-independent mechanism that might cause most of the type I HSRs to IV iron [7]. Iron–carbohydrate nanoparticles activate a complement [23] that generates C3a and C5a. Such anaphylatoxins bind specific mast cell receptors leading to the release of mediators including histamine, leukotrienes, prostaglandins and cytokines that trigger bronchospasm, edema, vasodilatation, muscle contraction and capillary leakage [24]. Iron reactions depend on the dosing and speed of infusion. Rapid IV iron administration is supposed to lead to a quick increase in labile free iron with activation of the complement system and quick accumulation of anaphylatoxins that leads to less efficient clearance of anaphylatoxins by carboxypeptidase N and other cells [9]. This process would worsen HSRs through the CARPA mechanism [24]. On the other hand, it is possible that some of the reactions to iron preparations may be caused by the transient presence of labile free iron (Table 1) in circulation yielded from the iron–carbohydrate complex (Figure 1) too quickly to be bound by transferrin. This may explain why less-stable iron–carbohydrate complexes, such as iron sodium gluconate, might induce non-IgE-mediated quick-onset reactions, particularly after a rapid high-dose infusion rate [22].

## 4. Clinical Manifestations

According to the time latency between the administration of IV iron and the onset of symptoms [9], HSRs can be classified as acute when they appear during the IV iron infusion that lasts about 30 min, especially within the first 10 min [18], or in 1 h after IV infusion [12,25], and less frequently, as delayed. Stojanovic et al. [25] reported that most reactions (184 out of 195; 94%) were immediate, while delayed reactions developed from 1 h to 5 days after the infusion in 11 children. Immediate HSRs include anaphylaxis and isolated immediate non-life-threatening symptoms, such as local irritation, rash, urticaria, angioedema, pruritus, nausea, vomiting, diarrhea, headache and myalgia [12,17,26,27,28,29].

In about 1 out of 100 patients [24,30], IV iron administration may elicit a Fishbane reaction. Though non-lethal, truncal muscle or joint aches, chest tightness and/or reddening of the face [31] occur in the absence of anaphylactic symptoms [32] or increase in serum tryptase level [33]. Symptoms quickly recover after stopping the infusion, and they do not recur when the infusion is restarted at a slower rate [31]. Fishbane reactions may be mistaken for prodromal symptoms of anaphylaxis [8]. Such reactions should not be treated with adrenaline or anti-H1 antihistamines, especially of the first generation (e.g., diphenhydramine and chlorpheniramine), as that can potentially provoke severe circulation inadequacy. Fishbane reactions have been hypothesized to be induced by labile iron rather than by CARPA [32]. 

The most common delayed reaction has been found to be cutaneous exanthema, which is usually considered to be T cell mediated [25]. Other delayed onset symptoms include fever, nausea, diarrhea, myalgia, arthralgia, lymphadenopathy or headache [32,34]. For a long time, it was unproved that iron was the trigger of delayed reactions. However, recently, Carron-Herrero et al. [34] have proved the onset of delayed reactions to iron using a drug provocation test (DPT).

## 5. Risk Assessment

If a patient reports a probable HSR after the administration of an iron product, it is of paramount importance to investigate their history: brand and generic name of the given product, route of administration, speed of infusion, clinical manifestations and chronology of reaction (first onset of symptoms, progression, time to resolution and response to treatment). The clinician may hypothesize the potential mechanism of the event and choose the appropriate tests, taking into account all these elements. Beside a detailed history, the allergy workup includes skin prick tests, intradermal test and DPT. A standardized extract for in vivo tests is not currently commercially available. Skin prick tests or intradermal test with iron solutions may lead to skin pigmentation that may long persist. In IgE-mediated reactions, also anecdotally reported against cow’s milk proteins in iron-compounds [35], skin tests are often positive due to the presence of IgE molecules bound to the mast cell surface, while in CARPA reactions, skin tests result negative. Nevertheless, skin tests’ diagnostic accuracy remains to be determined [36]. In Fishbane reactions, skin tests should not be performed. Patch tests may be useful for T-cell-mediated processes, such as delayed exanthemas, but, so far, optimal patch test concentrations have not been established. The determination of specific IgG to potential antigens or allergens in iron formulations, lymphocyte activation tests for delayed reactions and the basophil activation tests for immediate reactions are in vitro tests used for research purposes only. 

A case of delayed reaction (fever and asthenia occurring 24 h after administration) to iron carboxymaltose and iron sucrose has been confirmed by a positive lymphocyte activation tests 2 months after the reaction and a subsequent positive DPT [34]. A DPT is the only feasible way to identify an HSR to a drug irrespective of the involved mechanism. However, the DPT may elicit a life-threatening reaction; hence, it should be performed when the drug is necessary for the patient after a careful analysis of both risk and benefit [9]. A DPT may also be indicated to establish a firm diagnosis when facing a suggestive history of mild HSR with negative allergy tests [37] to identify patients who should be desensitized. 

Before starting a DPT, the child should be weighed, so that emergency drugs can be prepared. Then, incremental doses of the compound should be administered at established intervals, starting with low doses [38]. If no symptoms appear, the administration of the therapeutic daily dose of the tested drug should be achieved. DPT protocols widely vary between centers with respect to dose steps and time intervals [37]. To the best of our knowledge, no standardized protocol for a DPT to iron compounds is available so far. In order to achieve the daily dose, a protocol commonly used for DPT to beta-lactams, suggests starting with the administration of 5% of the full dose followed by 15%, 30% and 50% of the total amount, at 30 min intervals [39]. If a clinical history of anaphylaxis is present, the initial dose must be even lower (e.g., 1:10 or 1:100 of the usual first dose for DPT). Another protocol consists of giving 1/10, 2/10 and 7/10 of the therapeutic dose at 60′ intervals [38]. The patient remains under medical surveillance for one to two hours after the last dose. When discharged, the patient (or caregiver) is instructed to continue the observation period at home, for 48 h, and to treat delayed reactions appropriately [39].

## 6. Management of Intravenous Iron Infusions

IV iron should be administered under medical supervision by trained staff in settings where treatments and equipment for managing anaphylactic reactions are promptly available [18]. It seems worth reminding here that, when an HSR occurs, the infusion must be stopped. IM adrenaline represents the first option for anaphylaxis. Glucocorticoids are useful to avoid a protracted or biphasic course of anaphylaxis as well as respiratory and skin symptoms; antihistamines are helpful to treat reactions at the oral mucosa and skin reactions or oculorhinitis; a β2-agonist inhaler and oxygen can be required [7,24,40]. Before iron infusion, it is important to identify any contraindication [3,18].

Factors increasing the incidence or severity of HSRs should be documented. Unfortunately, information about risk factors for HSRs to IV iron formulations is limited. Atopic diseases, mild-to-moderate reaction to IV iron, HSRs to other drugs, high serum tryptase levels, asthma and urticaria are considered predisposing factors [8,24,32]. Generally, the severity of anaphylaxis to other IV drugs can be increased by male sex, older age, vigorous physical exercise, psychological distress, concomitant infections, comedications (non-steroidal anti-inflammatory drugs, beta-blocker or angiotensin converting enzyme inhibitor use), severe respiratory or cardiac disease or mastocytosis. Therefore, such conditions have been suggested to be linked to an increased severity of HSRs to IV iron [7,24,32]. In selected patients with risk factors for HSRs, preventive measures to minimize the risk of HSRs following IV iron administration are recommended. A preliminary small test dose is no longer recommended since it does not predict tolerance to the therapeutic dose [18]. Premedication, together with a slow infusion rate that is associated with lower rates of HSRs [32], and/or a lower dose following low reactogenic protocols (LRPs) should be implemented to reduce the risk of HSRs [7,18,24]. LRPs supported by the EMA [18] have been successfully employed for IV monoclonal antibodies [41] and liposomal drugs [36] whose HSRs may be caused by CARPA too [42]. When an LRP is adopted, a low-dose slow priming followed by a slow increasing of doses is also used over some days. The starting dose is low, from 0.001% to 0.01% of the full dose, and it is administered in 5–15 min. Premedication involves the parental administration of corticosteroid and/or antihistamines, even if no standardized protocol is available for both timing and doses [7]. In patients not at risk for HSRs to iron, premedication or LRP is not recommended. Premedication, especially with antihistamines, could mimic or mask some of the features of an HSR; therefore, it is a confounding variable when administered before IV iron [9].

When an HSR is ascertained through the analysis of all the available diagnostic tools and no alternative therapies are possible, a way to safely continue the iron administration is needed. Thus far, re-administration is not recommended for patients with HSRs to IV iron [24]. The EMA considers previous HSRs to IV iron a contraindication for its re-administration [18]. However, recent studies have shown that, after an infusion reaction, restarting the infusion at a slow infusion rate together with premedication under close surveillance was tolerated in patients with Fishbane reactions and mild or moderate HSRs to iron [25]. The same measures successfully prevent reactions when patients with a history of life-threatening reactions or delayed reactions are re-challenged to the same IV iron formulation or to an alternative one [25,43]. However, an immediate reaction to an alternative compound following re-challenge at a standard rate without premedication was reported in a patient with a previous moderate reaction [25]. Morales Matulena et al. [44] showed that only 3 out of 11 patients in whom the same IV iron product was re-administered developed an HSR during the re-administration. An alternative IV iron formulation with a slower infusion rate for 3 h was given to patients with a severe reaction (grade III or IV), and all of them were able to tolerate the product [44]. One patient presented with fever 2 days after the infusion. Controlled re-administration under close monitoring at slower infusion rates has also been suggested in a recent guideline to manage patients with previous HSRs to IV iron [24]. Re-exposure to the culprit product is not recommended. A different iron compound with slow infusion rates together with premedication may be tolerated even if it is not always safe [8,24,44,45]. 

Desensitization represents another possibility to induce tolerance of iron treatment. In the last decades, various case reports in adults have claimed its effectiveness for preventing treatment discontinuation and allowing therapeutic target achievement. Desensitization is a process through which a person with a known history of HSRs to a given stimulus becomes progressively less responsive by repetitive exposures [46]. Even though this tool is classically applied for type 1 allergy when a demonstrable or presumed IgE-mediated response is present, the same procedure has been used in other immunological reactions. The administration of increasing doses of the drug, by parenteral or oral routes can be used to carry out a desensitization process. The oral route is considered safer, although not always feasible. In these cases, the use of suspension formulations is preferred, since the desensitization protocol requires starting with very low doses that are easier to obtain with this form. Although no standardized protocol is currently available for iron desensitization, some desensitization schemes have been proposed for both oral and intravenous routes. An oral desensitization protocol, which was effective in two patients [46], started with the administration of 0.1 mL at concentration of 0.004 mg/mL of the commercial formulation of oral ferrous glycine sulfate. Increasing doses were given over three days up to the full dose: at 20 min intervals on the first day, while the following day they were administered at 1 h intervals. Patients tolerated the protocol and were then treated with the daily dose from the third day for 6 months, reaching the target values in their blood cell count at the end of the period. In 2017, a Colombian study group successfully performed a 10-step IV desensitization protocol in 2 patients. The protocol included premedication (systemic steroids, antihistamines and antileukotrienes), and it starts with an IV iron sucrose supply of 0.1 mg, with increasing graded doses until the 10th dose of 100 mg [47]. In 2019, an Italian group described three cases of effective desensitization in adults [36]. A 4-day oral desensitization protocol with the administration of sodium ferrogluconate was used in two patients, while the third underwent an IV desensitization protocol with 3 dilutions of ferric carboxymaltose (0.02, 0.2 and 2 mg/mL) administered at increasing concentrations, in 12 steps at 20 min intervals. The procedure was successfully repeated for the second time a week later. 

Allergen immunotherapy using the oral route has been shown to be of benefit in children with an allergy to inhalants and foods [48,49,50], but, so far, it has not been investigated in drug HSRs. Delayed reactions to iron are rare. One case developed fever and asthenia 24 h after the administration of IV iron carboxymaltose, and an alternative with iron sucrose caused fever and arthralgias after 4 h. The patient tolerated a rapid desensitization [8]. Severe cutaneous adverse reactions to iron have not been reported in children.

## 7. Conclusions

IV iron formulations are generally safe in children. Nevertheless, more knowledge has to be acquired to better prevent and manage HSRs to iron. New evidence in adults has shown that a slow infusion protocol with premedication or desensitization permits the administration of alternate iron in patients with iron HSRs. Avoidance of IV iron products [18] in the case of sparsely reported IV iron HSR may not be considered a standard practice. Although procedural guidelines have yet to be set, a tailored-medicine approach may be followed. Eventually, it is important to emphasize that these interventions in iron HSRs are yet to be better explored in children.

## Figures and Tables

**Figure 1 children-09-01473-f001:**
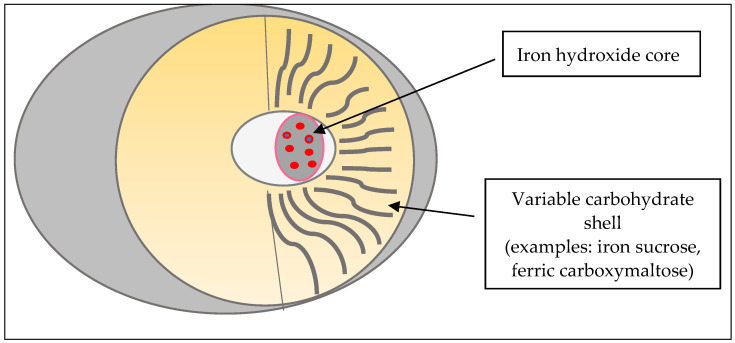
Iron–carbohydrate particles.

**Table 1 children-09-01473-t001:** Ligand [2], molecular weight [3], plasma half time [3] and rate of labile iron of compounds used for parenteral iron preparations.

Preparation	Ligand Description [2]	Molecular Weight (kD) [3]	Plasma Half Time (Hours) [3]	Labile Iron (% of Injected Dose) [3]
Second-generation iron products				
Iron sucrose	Disaccharide	30–60	6	3.5
Dextran (low-molecular-weight iron dextran)	Polysaccharide, maltose units 1→6-linked	165	20	2
Iron gluconate	Carboxylic acid and disaccharide	289–440	1	3.3
Third-generation iron products				
Ferric carboxymaltose	Polysaccharide, maltose units 1→4-linked	150	16	0.6
Ferumoxytol	Polysaccharide, maltose units 1→6-linked, hydrogenated and carboxymethylated	750	15	0.8
Iron isomaltoside	Oligosaccharide, maltose units 1→6-linked, hydrogenated and carboxylic acid	150	20	1

**Table 2 children-09-01473-t002:** Number of hypersensitivity reactions to intravenous iron formulations according to the U.S. Food and Drug Administration Adverse Event Reporting System database that included all of the population aged over 6 years [10] and incidence rate of anaphylaxis according to insurance claims from Medicare, U.S., in older people [11].

	Hypersensitivity Reaction (*n*) [10]	Anaphylactic Reaction (*n*) [10]	Deaths (*n*) [10]	Adjusted Incidence Rate of Anaphylaxis in 10,000 First Administrations [11]
Iron dextrans	57	22	4	9.8
Iron sucrose	212	43	7	1.2
Ferumoxytol	196	67	23	4
Ferric carboxymaltose	401	51	1	0.8

## Data Availability

Not applicable.

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
