# Peer review of "Hypersensitivity to Intravenous Iron Preparations"

_children, 2022, doi:10.3390/children9101473_

Round 1

Reviewer 1 Report

The manuscript is a narrative review of hypersensitivity reactions to IV preparations of iron, written by experienced clinicians from several children hospitals in Italy. The text is comprehensive, and covers all aspects of this clinical problem - etiology, epidemiology, pathophysiology, diagnostics, clinical picture, treatment, prevention and prognosis. The authors used relevant and recent medical literature as a basis of their statements. The topic of the manuscript is of great practical importance and summarize novel findings. The manuscript is written in good style and in very good English language. It us easy to read and main messages are clear and explicit. My only suggestion to the authors would be that a figure showing schematical structure of micelles with iron in the i.v. preparations may help the readers to understand the key issues even better.

Author Response

Answer:  We sincerely thank the referee for the comment on our paper. As requested, a new figure on micelles with iron has been included. 

Reviewer 2 Report

This paper mainly discussed about the prevalence, mechanism, symptoms, diagnosis and management of hypersensitivity related to IV iron infusions. Overall the content was well organized. However, I noticed some issues that need to be addressed.

1. The paper was submitted to the journal Children, but about half of the literatures it included did not specifically focus on children or even young adults. I'm not sure if it's out of the journal's scope.

2. Regarding to the content, tables need to be updated. 

   a) In table 2, the numbers for column 1,2,3 were from a paper, based on FDA dataset, which included all-age population. However, the column 4 was from a paper among "American older adults'. Plus, numbers in the column 4 were mismatched with the iron formulations.

   b) I don't feel the contents in table 3 need to be placed in a table. If the authors insisted to do so, please consider other table format, or at least remove all the bullets.

3. In line 98, please confirm if the 52 cases means 52 patients, or 52 doses.

4. It is quite confused between line 96-104. Authors talked about data from 3 different publications, but it's hard to tell which result came from which paper. Please re-write this part.

5. Section 5 is more likely to be "risk assessment" rather than "diagnosis".

6. Section 7 is also a part of management. It should not be listed as a independent section.

Author Response

This paper mainly discussed about the prevalence, mechanism, symptoms, diagnosis and management of hypersensitivity related to IV iron infusions. Overall, the content was well organized. However, I noticed some issues that need to be addressed.

  1. The paper was submitted to the journal Children, but about half of the literatures it included did not specifically focus on children or even young adults. I'm not sure if it's out of the journal's scope.

Answer.  Iron hypersensitivity is an important issue in children. Regrettably, there is a paucity of studies on this crucial point in pediatric age and many series reported in the literature include both adults and children. So, we were forced to consider studies both in adults and children. However, we believe that our work assessing the most relevant publications from the perspective of the pediatrician, is helpful for those who take care of people in the pediatric age. 

  1. Regarding to the content, tables need to be updated. 
  2.  a) In table 2, the numbers for column 1,2,3 were from a paper, based on FDA dataset, which included all-age population. However, the column 4 was from a paper among "American older adults'. Plus, numbers in the column 4 were mismatched with the iron formulations.

Answer. Table 2  has been changed.

  1.  b) I don't feel the contents in table 3 need to be placed in a table. If the authors insisted to do so, please consider other table format, or at least remove all the bullets.

Answer. According to the reviewer’s request, Table 3 has been deleted.

  1. In line 98, please confirm if the 52 cases means 52 patients, or 52 doses.

Answer. We have now clearly stated that there were 52 patients.

  1. It is quite confused between line 96-104. Authors talked about data from 3 different publications, but it's hard to tell which result came from which paper. Please re-write this part.

Answer. Thank you for comment. Reference numbers were not in the appropriate place in the text. We have changed them.

  1. Section 5 is more likely to be "risk assessment" rather than "diagnosis".

Answer. According to the reviewer’s request, the subtitle has been changed.

  1. Section 7 is also a part of management. It should not be listed as a independent section.

Answer. According to the reviewer’s request, the subtitle has been erased.

Round 2

Reviewer 2 Report

Appreciate the authors quick response and all the efforts to modify their work.